# OpenReview forum: "Contrastive Order Learning for Ordinal Regression"
_ICLR.cc/2026/Conference — ICLR 2026 Conference Withdrawn Submission_

### Official Review · Reviewer_nC2o · 2025-11-01

**Soundness:** 3
**Presentation:** 3
**Contribution:** 3
**Rating:** 6
**Confidence:** 2

**Summary:**

This paper introduces a novel algorithm for ordinal regression called Contrastive Order Learning (ConOrd), which aims to combine the strengths of contrastive learning and order learning. Its core contribution is a contrastive order loss, which uses "soft" affinity and disparity weights, enables comparison among all sample pairs within a batch. The authors validate ConOrd through extensive experiments on diverse ordinal regression tasks, including facial age estimation, blind image quality assessment, and blind video quality assessment.

**Strengths:**

1 Clear Motivation and Problem Formulation. The paper is well-motivated and clearly articulates the respective limitations of standard contrastive learning (ignores order) and traditional order learning (local comparisons, limited batch exploitation). The proposed ConOrd method is a logical and well-designed solution to bridge this gap.
2 Novel and Intuitive Loss Function. The primary strength is the novel ConOrd loss. The use of soft affinity and disparity weights based on rank differences is an elegant and effective mechanism. It intuitively captures the ordinal structure by applying fine-grained attraction and repulsion forces across all pairs, which is a significant improvement over methods like RnC that use hard, binary rank-based thresholds.
3 Exceptional Empirical Performance and Generalizability. The extensive experimental validation covers a comprehensive range of tasks. The method's effectiveness is demonstrated by its outperformance across a wide array of benchmarks, from age estimation to perceptual quality scoring and physical measurements. This strongly supports the algorithm's generalizability.

**Weaknesses:**

1 Limited Scope of Contribution: The paper's core technical contribution is a new loss function, $L_{ConOrd}$. As clearly illustrated in Figure 1, this approach is designed to overcome the limitations of standard supervised contrastive learning (SupCon), which ignores ordinal relationships, and existing ordinal contrastive learning methods like RnC, which use a "hard-assigned" sample selection strategy that potentially overlooks valuable training pairs. ConOrd's use of soft-weighting to incorporate all pairs in a batch is a notable and intuitive refinement. However, restricting the primary contribution solely to an improved loss function raises concerns about the scope and breadth of the work. The work currently presents as a valuable but potentially incremental improvement to the contrastive learning paradigm for ordinal tasks.
2 Tenuous Connection to the Order Learning Field: The paper positions itself as "unifying the strengths of order learning (OL) and contrastive learning (CL)." However, this connection to the established order learning literature appears weak and superficial. The method's derivation, as presented in the motivation (Section 3.2), begins by analyzing the shortcomings of a standard contrastive learning method when applied to ordinal tasks. The entire approach feels more like 'injecting ordinal-awareness into contrastive learning' rather than a deep, principled fusion with the foundational theories and methods of ordinal learning.
3 Unclear Contribution of $L_{center}$: The paper's core method combines $L_{ConOrd}$ with $L_{center}$, making it difficult to isolate the true performance contribution of $L_{ConOrd}$ itself. The ablation in Table 5 compares $L_{ConOrd} + L_{center}$ against a baseline $L_{RnC}$ , but lacks an experiment evaluating $L_{RnC} + L_{center}$. If this combination also significantly outperforms the $L_{RnC}$ baseline, it would suggest that the $L_{center}$ regularization is a primary driver of the performance gains. To properly disentangle these effects and validate the superiority of the proposed loss, the authors should provide a more comprehensive ablation, ideally comparing $L_{SupCon}$, $L_{RnC}$, and $L_{ConOrd}$ both with and without the $L_{center}$ component.

**Questions:**

1 Could you please elaborate on the connections between the proposed method and Order Learning (OL)? Specifically, how is the proposed method rooted in OL, or what direct inspirations were drawn from foundational OL theories?
2 Please provide a more comprehensive ablation comparing $L_{SupCon}$, $L_{RnC}$, and $L_{ConOrd}$ both with and without the $L_{center}$ component.

---

> ### Author Response · Authors · 2025-11-19
> **We thank the reviewer for the thoughtful and constructive feedback.**
>
> Thank you for your positive review. Please find our responses below, all of which have also been incorporated into the revised manuscript, with newly added content highlighted in blue.
>
> ***
> > **Effect of $L_\text{center}$**
>
> We appreciate the reviewer’s concern regarding the effect of the center-loss regularization. To isolate its contribution, we performed an extended ablation comparing $L_{\text{SupCon}}$, $L_{\text{RnC}}$, and $L_{\text{ConOrd}}$, both with and without the center loss.
>
> |Loss|CLAP2015|BID|BID|LSVQ-1080p|LSVQ-1080p|
> |-|-|-|-|-|-|
> ||MAE ($\downarrow$)|SRCC ($\uparrow$)|PCC ($\uparrow$)|SRCC ($\uparrow$)|PCC ($\uparrow$)|
> |$L_{\text{SupCon}}$|2.625|0.819|0.876|0.614|0.682|
> |$L_{\text{SupCon}}+L_{\text{center}}$|2.610|0.815|0.875|0.622|0.689|
> |$L_{\text{RnC}}$|2.531|0.892|0.906|0.812|0.843|
> |$L_{\text{RnC}}+L_{\text{center}}$|2.745|0.892|0.906|0.808|0.839|
> |$L_{\text{ConOrd}}$|2.509|0.909|0.925|0.815|0.848|
> |$L_{\text{ConOrd}}+L_{\text{center}}$|2.461|0.913|0.925|0.818|0.851|
>
> The results show that $L_{\text{ConOrd}}$ already surpasses $L_{\text{SupCon}}$ and $L_{\text{RnC}}$, even without the center term, confirming that its effectiveness stems from the loss formulation itself. Adding the center term yields only minor gains and does not affect the relative ordering among methods; in particular, $L_{\text{RnC}} + L_{\text{center}}$ does not improve over $L_{\text{RnC}}$. Please refer to Table 15 on page 23 of the revised manuscript.
> ***
> > **Connection to Order Learning**
>
> We appreciate the reviewer’s request for a clearer explanation of how ConOrd relates to the order learning (OL) literature. For completeness, we briefly summarize the connection here and refer the reviewer to Section 3.2 of the revised manuscript, where these links are now described more explicitly.
>
> ConOrd draws on three principles established in OL. Lim et al. (2020) emphasize that relative comparisons provide more reliable supervision than absolute predictions, which motivates ConOrd’s use of all pairwise relations in a batch. Shin et al. (2022) extend this idea by showing that the magnitude of the rank difference itself can be used as informative supervision, motivating our use of continuous weights based on rank differences. Lee et al. (2022) introduce attraction-repulsion forces to shape the embedding geometry according to ordinal structure; ConOrd builds on this idea by applying smoothly weighted attraction and repulsion to all pairs.
>
> These principles form the basis for the smooth, fully pairwise ordinal contrastive formulation in ConOrd. The revised manuscript clarifies this connection in Section 3.2 (Lines 192–209).
>
>
> ***
> > **Summary**
>
> The revised manuscript incorporates all suggestions raised by the reviewer, including:
> - **analysis on the effect of $L_\text{center}$ on competing losses**,
> - **clarified connection between ConOrd and established order learning methods**.
>
> We sincerely thank the reviewer for the detailed comments. They were particularly helpful in clarifying and sharpening the motivation behind ConOrd, and they improved the presentation of the revised manuscript. We would be glad to address any further questions or provide additional analysis if needed.

---

### Official Review · Reviewer_SiKZ · 2025-11-01

**Soundness:** 3
**Presentation:** 3
**Contribution:** 3
**Rating:** 4
**Confidence:** 5

**Summary:**

The paper proposes a novel contrastive order learning algorithm, ConOrd, designed for the challenging task of Ordinal Regression. The method cleverly integrates the benefits of supervised contrastive learning with fine-grained ordinal constraints. This is achieved by introducing rank difference-based affinity ($a_{ij}$) and disparity ($b_{ij}$) weights to precisely model the ordinal relationships among all samples within a batch, promoting better rank alignment and clustering in the embedding space. Furthermore, the authors introduce a Center Loss ($L_{\text{center}}$) to regularize the embedding space by pulling samples of the same rank toward their respective centers. Extensive experiments on diverse ordinal regression tasks, show that ConOrd consistently outperforms existing state-of-the-art methods, demonstrating its effectiveness and generalization ability. However, while the empirical results are strong, the paper needs more rigorous theoretical grounding and a more comprehensive set of baseline comparisons to fully justify its novelty and complexity.

**Strengths:**

1. The ConOrd method consistently and stably achieves state-of-the-art results across multiple benchmark datasets, including MORPH II, CLAP2015, LSQV-1080P, and KonVid-1k. Its superior performance is particularly evident in the BIQA and BVQA tasks.

2. The core innovation lies in the $L_{\text{ConOrd}}$ loss, which skillfully combines contrastive learning with ordinal structure via fine-grained, rank-aware weighting. The design of the affinity and disparity weights ($a_{ij}, b_{ij}$) that utilize continuous rank differences is an elegant solution compared to methods relying only on binary pairs or fixed margins.

3. The model's efficacy is demonstrated across three distinct ordinal regression domains (age estimation, image quality, and video quality), underscoring its versatility and strong generalization capacity.

4. The inclusion of the Center Loss ($L_{\text{center}}$) is shown in ablation studies (Figure 6, Table 6) to be crucial for the final performance, as it effectively regularizes the embedding space to achieve tighter clustering for samples belonging to the same rank.

**Weaknesses:**

1. The paper lacks theoretical analysis or proof of convergence for the proposed $L_{\text{ConOrd}}$ loss function. Given that contrastive losses can be sensitive to hyperparameter choices and sampling, providing theoretical insights into its stability, gradient properties, or generalization bounds is crucial, yet missing.

2. While comparisons are made against many existing ordinal regression methods, the paper lacks a comprehensive and fair comparison against the latest Transformer-based or large-scale pre-trained model baselines. For instance, a side-by-side comparison using the same ViT-B backbone (only mentioned briefly on page 7) across all tasks against SOTA methods is absent, reducing the overall persuasiveness of the performance gains.

3. The $L_{\text{ConOrd}}$ loss relies on several critical hyperparameters ($\epsilon, \gamma, \alpha_0, \beta_0$). The ablation study (Table 7, page 9) only shows 8 variations of the loss structure. A thorough sensitivity analysis (ideally a visual plot) for key parameters like $\epsilon$ and $\gamma$ is necessary to understand their impact on final performance and training stability.

4. The specific functional forms used for the affinity and disparity weights (Equations 4 and 5), such as $(|r_i - r_j| + \epsilon)^{-1}$, are clever but their mathematical derivation and justification lack depth. The authors should provide a clearer, intuitive argument for why these specific inverse distance formulations are superior to other potential weighting schemes.

**Questions:**

1. Regarding the $L_{\text{ConOrd}}$ loss function, how did the design of the weights $a_{ij}$ and $b_{ij}$ ensure effective suppression of gradient explosion and maintain training stability? Please provide a more in-depth discussion, either through theoretical arguments or analysis of the gradient dynamics.

2. In Table 7, you present 8 alternative configurations for $L_{\text{ConOrd}}$. Could you provide a dedicated sensitivity analysis plot for the key hyperparameters (e.g., $\epsilon$ and $\gamma$) to visually demonstrate their influence on the final performance and convergence speed?

3. Given the importance of $L_{\text{center}}$, please discuss the impact of the initialization method of the center points $\mu_m$ on the final results. What is the observed difference in performance if the centers were initialized randomly instead of being pre-calculated based on the training set?

4. Whether the proposed method can provide meaningful visual explanations, such as grad-cam?

---

> ### Author Response · Authors · 2025-11-20
> **We appreciate the reviewer’s thorough assessment and helpful suggestions (Part I).**
>
> Thank you for your constructive comments. Please find our responses below, all of which have also been incorporated into the revised manuscript, with newly added content highlighted in blue.
>
> ***
> > **Stability, gradient behavior, and justification of the weighting design**
>
> We appreciate the reviewer’s detailed questions. Section 3.3 (p.5) and Appendix A (p.16) of the revised manuscript now provide an expanded discussion, summarized below.
>
> (1) Gradient behavior and stability.
>
> The gradient analysis in Appendix A shows that each pairwise update takes the form
>
> $$
> \exp(\kappa_{ij}/\tau)\Big(\tfrac{a_{ij}}{\alpha_i}-\tfrac{b_{ij}}{\beta_i}\Big)(z_i - z_j),
> $$
>
> where the soft-normalization terms $\alpha_i$ and $\beta_i$ ensure that no single pair dominates the update. Because the affinity $a_{ij}$ decreases smoothly with the rank gap and the disparity $b_{ij}$ increases smoothly, both attraction and repulsion components remain well-scaled throughout training. Empirically, all eight variants in Table 7 (p.9) yield smooth loss curves and bounded gradient norms (Figure 15, p.26), confirming that the weighting scheme avoids gradient explosion in practice.
>
> (2) Intuition for the quadratic affinity-disparity weights.
>
> The choices
>
> $$
> a_{ij}=\tfrac{1}{d_{ij}^{2}+\epsilon}, \qquad b_{ij}=d_{ij}^{2}
> $$
>
> satisfy three requirements identified in the gradient analysis:
> - **Ordinal consistency:** smoothly weakening attraction for larger rank gaps and smoothly strengthening repulsion.
> - **Gradient stability:** quadratic functions are second-order smooth and avoid abrupt changes that destabilize optimization.
> - **Robustness:** Tables 7 (p.9) and 14 (p.23) show that several monotonic weighting forms behave well, with method VIII consistently performing strongly. This indicates that method VIII is a natural, stable instantiation of the framework rather than a finely tuned or fragile choice.
>
> (3) Scope of theoretical analysis.
>
> While we do not provide a formal convergence proof — consistent with prior contrastive-learning literature — we do offer explicit gradient expressions, analyze the role of rank-dependent weights, and empirically verify stable training dynamics across all design variants.
>
> ***
> > **Comparison with transformer-based baselines**
>
> We appreciate the reviewer’s question regarding comparisons with recent transformer-based methods. Our main results (Tables 2–4) already include SOTA baselines that use the same ViT-B backbone as ConOrd, ensuring a fair and capacity-matched evaluation. To make this clearer in the revision, we added a “Backbone” column to Tables 2–4.
>
> In addition, Table 5 provides a fully controlled comparison in which all models share the same ViT-B backbone and differ only in their loss functions, isolating the contribution of ConOrd itself. We hope these clarifications address the reviewer’s concern and enhance the transparency of our evaluation.
>
> ***
> > **Hyperparameter analysis**
>
> We would like to clarify that the proposed loss contains only two hyperparameters during training: the temperature parameter $\tau$ and the affinity-weight parameter $\epsilon$. The eight variants in Table 7 correspond to different structural choices of the loss (combinations of similarity, affinity, and disparity functions), not hyperparameter settings. The actual hyperparameters $\tau$ and $\epsilon$ were already evaluated numerically in Tables 10 and 11 on p.21. For convenience, we reproduce the corresponding BID results below.
>
> |$\tau$|SRCC↑|PCC↑|
> |-|-|-|
> |0.05|0.910|0.925|
> |0.06|0.912|0.926|
> |0.07|0.913|0.925|
> |0.08|0.913|0.931|
> |0.09|0.911|0.921|
> |0.10|0.911|0.924|
> |0.50|0.901|0.913|
> |1.00|0.888|0.905|
> |1.50|0.883|0.901|
> |2.00|0.888|0.900|
>
> |$\epsilon$|SRCC↑|PCC↑|
> |-|-|-|
> |$10^{-3}$|0.915|0.928|
> |$10^{-4}$|0.925|0.926|
> |$10^{-5}$|0.914|0.929|
> |$10^{-6}$|0.914|0.930|
> |$10^{-7}$|0.921|0.925|
> |$10^{-8}$|0.910|0.928|
> |$10^{-9}$|0.911|0.925|
>
> These numerical results already show that ConOrd is stable across a broad range of  $\tau$ and $\epsilon$ values. In response to the reviewer’s suggestion, we additionally included dedicated sensitivity plots that visualize the effect of each hyperparameter on both final performance (SRCC / PCC) and training dynamics. These analyses are now provided in Figure 10 (p.21) for $\tau$ and Figure 11 (p.22) for $\epsilon$.
>
> ***
> *(Continued in Part II)*

---

> ### Author Response · Authors · 2025-11-20
> **We appreciate the reviewer’s thorough assessment and helpful suggestions (Part II).**
>
> *(Continued from Part I)*
>
> ***
> > **Initialization of $\mu_m$**
>
> In our setup, the center points $\mu_m$ are initialized randomly. To evaluate the effect of this choice, we compared several common initialization schemes on the BID dataset — random, zeros, truncated normal, and Kaiming normal. All four exhibit nearly identical performance.
>
> | Init. method | Random | Zeros | Trunc. Normal | Kaiming Normal |
> |---|---|---|---|---|
> | SRCC | 0.913 | 0.910 | 0.913 | 0.910 |
> | PCC  | 0.925 | 0.922 | 0.925 | 0.925 |
>
> We also tested the reviewer’s suggestion of "pre-computing centers from the mean feature of each rank." This yields only a marginal improvement but requires a full pass over the training set, making it far more expensive.
>
> | Method | SRCC | PCC | Time |
> |---|---|---|---|
> | Random init. | 0.913 | 0.925 | 1.21×10⁻³ s |
> | Mean-feature init. | 0.915 | 0.932 | 5.25 s |
>
> Given the modest benefit relative to the substantial overhead, random initialization remains the most practical and efficient choice. This has been added to Appendix C.3 on page 23 (Tables 16&17).
>
> ***
> > **Grad-CAM visualization**
>
> As suggested by the reviewer, we applied Grad-CAM to the encoder for the BIQA task and generated visual explanation maps. The resulting visualizations show that ConOrd tends to activate semantically meaningful regions that correlate with perceptual quality. These Grad-CAM maps have been added as Figure 14 on page 25 of the revised manuscript.
>
>
> ***
> > **Summary**
>
> We have carefully incorporated the reviewer’s insightful suggestions throughout the revised manuscript, including:
> - **more detailed analysis of the weighting schemes and gradient behavior**,
> - **clearer comparisons against baselines using identical ViT-B backbones**,
> - **visual sensitivity studies for the key hyperparameters**,
> - **comparative evaluations of center initialization strategies**,  and
> - **Grad-CAM–based qualitative visualizations**.
>
> Your feedback has significantly improved both the technical rigor and the clarity of our presentation. We would be glad to provide any further explanation or additional analysis if needed.

---

### Official Review · Reviewer_F4NJ · 2025-11-02

**Soundness:** 3
**Presentation:** 4
**Contribution:** 2
**Rating:** 4
**Confidence:** 4

**Summary:**

This paper presents Contrastive Order Learning (ConOrd), a representation-learning framework for ordinal regression problems. ConOrd incorporates (1) a contrastive order loss that applies soft affinity and disparity weights derived from rank differences, and (2) a center loss that encourages compact feature clustering. Extensive experiments across multiple ordinal regression benchmarks demonstrate that ConOrd consistently achieves state-of-the-art performance.

**Strengths:**

+ The proposed method works well on multiple benchmarks.
+ The proposed loss is plug-and-play, which can be integrated with other backbones.
+ It excels at various modalities.

**Weaknesses:**

+ Methodological novelty is somewhat limited as it mainly combines constrative learning and ordinal learning. The center loss is adopted from the literature.
+ During the inference stage, the test-time k-NN may not be stable.
+ Statistical analysis may be needed as there is only average performance.

**Questions:**

1. I am curious about the effectiveness of the test-time k-nn and the effect of k.
2. More informative comparisons, such as standard deviation and computational cost, could be reported.
3. The author should justify the reason for the L_conord configuration. Any theoretical consideration?

---

> ### Author Response · Authors · 2025-11-19
> **We appreciate the reviewer’s careful evaluation and thoughtful feedback (Part I).**
>
> Thank you for your constructive comments. Please find our responses below; all suggestions have been incorporated into the revised manuscript, with newly added or modified content highlighted in blue.
> ***
> > **Test-time k-NN and sensitivity to $k$**
>
> ConOrd performs test-time $k$-NN regression using pre-computed training embeddings, resulting in lightweight and efficient inference. To evaluate its stability, we measured the total evaluation time over 10 independent runs and report the mean and standard deviation below. As shown, ConOrd achieves faster inference across all benchmarks and exhibits smaller variance, indicating stable test-time behavior. These results are included in Table 22 on p.27 of the revised manuscript.
>
> |Method|CLAP2015 (Age)|BID (BIQA)|LSVQ-test (BVQA)|
> |-|-|-|-|
> |SOTA baseline|NumCLIP: $5.15\pm0.42$s|LoDa: $5.63\pm0.31$s|DOVER: $1222.36\pm9.04$s|
> |ConOrd ($k$-NN)|$2.16\pm0.14$s|$0.88\pm0.05$s|$253.70\pm7.65$s|
>
> To further examine the stability of $k$-NN inference, we varied $k$ during test time and evaluated MAE (age task) or SRCC/PCC (quality tasks). The results are summarized below.
>
> |$k$|CLAP2015|AgeDB|BID|BID|LSVQ-test|LSVQ-test|
> |-|-|-|-|-|-|-|
> ||MAE($\downarrow$)|MAE($\downarrow$)|SRCC ($\uparrow$) | PCC ($\uparrow$) | SRCC ($\uparrow$) | PCC ($\uparrow$) |
> |4|2.461|5.232|0.910|0.924|0.894|0.892|
> |10|2.483|5.159|0.913|0.925|0.901|0.900|
> |20|2.496|5.158|0.909|0.923|0.903|0.903|
> |30|2.503|5.158|0.910|0.924|0.904|0.904|
> |40|2.496|5.154|0.910|0.923|0.904|0.904|
> |50|2.487|5.158|0.910|0.923|0.904|0.904|
> |60|2.486|5.145|0.910|0.923|0.904|0.904|
>
> CLAP2015 — being a relatively small dataset — exhibits mild fluctuations as $k$ varies. In contrast, for larger datasets such as AgeDB, BID, and LSVQ, the performance becomes stable once $k$ exceeds a modest threshold, showing almost no sensitivity to the choice of $k$. Following this observation, we use $k=4$ for small age datasets (MORPH, CLAP) and $k=60$ for larger ones, while adopting $k=10$ for BIQA and $k=30$ for BVQA to match dataset scale and ensure stable accuracy. These results have been added to Table 12 on page 22 of the revised manuscript.
>
> ***
> > **Standard deviation of performance**
>
> As suggested by the reviewer, we conducted additional experiments using multiple random seeds and report the mean and standard deviation across runs to assess the stability of each method. Specifically, for age estimation (CLAP2015) and BVQA (LSVQ-test), we used 5 random seeds, while for the BIQA (BID) task we used 10 seeds. The results are summarized below and have been incorporated into Table 20 on page 25 of the revised manuscript.
>
> |Method|CLAP2015|BID|BID|LSVQ-test|LSVQ-test|
> |-|-|-|-|-|-|
> ||MAE($\downarrow$)|SRCC($\uparrow$)|PCC($\uparrow$)|SRCC($\uparrow$)|PCC($\uparrow$)|
> |$L_{\text{SupCon}}$|2.6058$\pm$0.0121|0.8182$\pm$0.0052|0.8754$\pm$0.0035|0.7576$\pm$0.0034|0.7660$\pm$0.0032|
> |$L_{\text{RnC}}$|2.5324$\pm$0.0202|0.8901$\pm$0.0260|0.9041$\pm$0.0237|0.9024$\pm$0.0006|0.9004$\pm$0.0011|
> |ConOrd|2.4698$\pm$0.0122|0.9118$\pm$0.0243|0.9255$\pm$0.0183|0.9040$\pm$0.0007|0.9036$\pm$0.0009|
>
> Across all benchmarks, ConOrd not only achieves higher average performance than RnC but also consistently exhibits smaller variance across runs, indicating more stable optimization and reduced sensitivity to random initialization.
>
> ***
> ***
> > **Computational cost**
>
> We appreciate the reviewer’s request for a more detailed analysis of computational efficiency. Overall training and inference costs were already reported in Appendix C.4 on page 26. To further clarify the efficiency of the proposed loss itself, we additionally provided a fine-grained comparison of the loss-level computation time for $L_{\text{ConOrd}}$, $L_{\text{RnC}}$, and the SOTA BIQA loss from QCN.
>
> | |$L_{\text{ConOrd}}$|$L_{\text{RnC}}$|QCN|
> |-|-|-|-|
> |Loss computation time (batch size = 32)|4.6 ms|8.7 ms|8.5 ms|
>
> We also compared GPU memory usage required solely for computing each loss:
>
> |Algorithm|Memory|
> |-|-|
> |$L_{\text{ConOrd}}$|2.17 MB|
> |$L_{\text{RnC}}$|1.17 MB|
> |QCN|273.78 MB|
>
> Thus, $L_{\text{ConOrd}}$ is more efficient in both time and memory than $L_{\text{RnC}}$ and the QCN loss. These results have been added to Tables 23 and 24 on page 27 of the revised manuscript.
>
> ***
> *(Continued in Part II)*

---

> ### Author Response · Authors · 2025-11-20
> **We appreciate the reviewer’s careful evaluation and thoughtful feedback (Part II).**
>
> *(Continued from Part I)*
>
> ***
> > **Justification for the $L_{\text{ConOrd}}$ configuration**
>
> The choice of affinity and disparity weights in $L_{\text{ConOrd}}$ is guided by three requirements derived from the theoretical analysis in Section 3.3 (p. 5) and Appendix A (p. 16).
> (1) **Ordinal consistency:** The gradient analysis shows how the embeddings attract or repel each other according to their rank gap; using quadratic affinity and disparity weights naturally yields smoothly varying attraction forces for nearby ranks and repulsion forces for distant ranks, which align with the desired ordinal behavior.
> (2) **Gradient stability:** Quadratic forms are second-order smooth and yield bounded, well-behaved gradients, as confirmed in the training-stability analysis (Figure 15, p. 26).
> (3) **Robustness across variants:** Ablations in Table 7 (p. 9) and Table 14 (p. 23) show that ConOrd is stable across many choices of affinity-disparity weights, with method VIII consistently performing well; it therefore represents a natural, well-behaved instantiation of the framework rather than a finely tuned or fragile choice.
>
> ***
> > **Summary**
> We have thoroughly addressed the reviewer’s recommendations in the revised manuscript, including:
> - **test-time $k$-NN behavior and the effect of $k$**,
> - **standard deviation analysis across multiple seeds**,
> - **additional comparisons of training and inference cost**,
> - **justification for the $L_{\text{ConOrd}}$ configuration based on the gradient analysis**.
>
> We sincerely thank the reviewer for the constructive feedback, which helped improve the rigor and clarity of our work. We would be glad to provide any further clarification or additional analysis if needed.

---

### Official Review · Reviewer_jdzb · 2025-11-02

**Soundness:** 3
**Presentation:** 3
**Contribution:** 3
**Rating:** 8
**Confidence:** 4

**Summary:**

A method is proposed to model ordinal regression using contrastive learning.  The key innovation is modifying the InfoNCE loss to include two factors: 1) the affinity of the two samples, and 2) the distance between the two samples.   The idea is to minimize the expected rank estimation error with respect to the contrastively learned probability distribution.  A key advantage is that all the samples in the rank order can be used, without needing to separate them into positive and negative groups, as in current contrastive learning methods.

The second innovation is to introduce a center loss, which requires all samples within rank m to be closer to each other in the contrastively learned image feature space.

Experiments on the standard dataset show that this method works well.

**Strengths:**

This paper introduced a simple but effective solution for extending contrastive learning to rank regression.  The idea and implementation of minimizing the expected rank estimation error with respect to the contrastively learned probability distribution are good ones.  The centering loss is a simple addition, and it is shown to be very effective.

Together, the method simplifies many ad hoc designs, where one has to specify positive vs negative pairings in the contrastive learning.  The centering loss might replace the `momentum contrastive' implementation, which is used in all current contrastive learning methods.

The paper is well presented, and the figures are precise.

**Weaknesses:**

In practical applications, absolute ordering training data are often difficult to obtain and may contain errors.  It would be helpful to address how robust this method is to incomplete ordering information or to errors in training data.   This is a concern since this method employs a disparity measure that assigns large weights to distant pairs.

A more detailed comparison to Khosla et al., 2020, RnC algorithm (Zha et al., 2023), in terms of the embedding features learned would be helpful.

**Questions:**

See above weakness.

---

> ### Author Response · Authors · 2025-11-19
> **We appreciate the reviewer’s positive evaluation and constructive questions.**
>
> Thank you for your encouraging review. Please find our responses below, all of which have also been incorporated into the revised manuscript, with newly added content highlighted in blue.
>
> ***
>
> > **Robustness to incomplete ordering information and data corruption**
>
> To evaluate robustness, we reproduced the two experimental protocols used in the RnC paper — resilience to reduced training data and robustness to data corruptions — using the AgeDB dataset. We compared $L_{\text{SupCon}}$, $L_{\text{RnC}}$, and our proposed ConOrd under identical settings.
>
> (1) Resilience to reduced training data
>
> To examine robustness to incomplete ordering information, we progressively subsampled the training set. The table below shows the MAE performance at each sampling ratio, where ConOrd consistently surpasses both SupCon and RnC. The performance gap becomes more pronounced in the low-data regime (e.g., 0.1 and 0.3 ratios), confirming that ConOrd learns more sample-efficient and stable ordinal representations when ordering information is scarce.
>
> |Sampling ratio|$L_{\text{SupCon}}$|$L_{\text{RnC}}$|$L_{\text{ConOrd}}+L_{\text{center}}$|
> |-|-|-|-|
> |0.1|6.948|6.953|6.690|
> |0.3|6.022|6.009|5.881|
> |0.5|5.855|5.655|5.574|
> |0.8|5.457|5.287|5.261|
> |1.0|5.361|5.192|5.145|
>
> (2) Robustness to data corruptions
>
> Following the ImageNet-C corruption protocol [1], we applied 19 corruption types at severity levels 0–5 to the test data while training all models on clean images. As shown in the table below, ConOrd achieves the best performance at the clean level (severity 0) and exhibits markedly slower degradation as corruption severity increases. Even under the strongest corruption (severity 5), ConOrd maintains substantially lower MAE (10.187) than SupCon and RnC, demonstrating superior robustness to distribution shift and corrupted inputs.
>
> |Corruption severity level|$L_{\text{SupCon}}$|$L_{\text{RnC}}$|$L_{\text{ConOrd}}+L_{\text{center}}$|
> |-|-|-|-|
> |0|2.625|2.531|2.461|
> |1|6.375|6.253|5.993|
> |2|7.267|7.127|6.718|
> |3|8.079|7.957|7.430|
> |4|9.399|9.311|8.589|
> |5|11.204|11.165|10.187|
>
> [1] Dan Hendrycks and Thomas Dietterich. "Benchmarking neural network robustness to common corruptions and perturbations." ICLR 2019
>
> These results have been incorporated into Tables 18 and 19, and the corresponding performance trends are visualized in Figure 12 on page 24 of the revised manuscript.
>
> ***
> > **More comparison to RnC**
>
> We appreciate the reviewer’s suggestion regarding a deeper comparison with RnC. In response, we have added a new t-SNE visualization in the revision (page 25, Figure 13), directly comparing the embedding structures learned by RnC and ConOrd. The updated figure highlights that ConOrd forms a more coherent and smoothly ordered embedding space. This additional analysis strengthens the empirical distinction between the two methods.
>
> ***
> > **Summary**
>
> We have carefully incorporated the reviewer’s recommendations into the revised manuscript, including:
> - **robustness evaluations under reduced training information and data corruptions**, and
> - **embedding-level comparisons with RnC**.
>
> We sincerely appreciate the reviewer’s thoughtful feedback, which has helped improve the clarity and presentation of our work. Please let us know if any further clarification or additional analysis would be helpful.

---

### Author Response · Authors · 2025-11-12

We would like to thank all reviewers for dedicating their time and effort to providing valuable feedback. We will share our responses to each question or comment as soon as possible.

---

### Note · Authors · 2026-01-27

I have read and agree with the venue's withdrawal policy on behalf of myself and my co-authors.

---

### Meta-Review · Area_Chair_Ef8T · 2026-01-06

**Summary:**

The paper presents an empirical framework for ordinal regression, but reviewers raised significant concerns regarding limited methodological novelty. The core contribution is seen as an incremental refinement of existing contrastive losses, lacking the theoretical rigor or foundational innovation required for a higher-tier acceptance.

**Reviewer Concerns:**

The rebuttal clarified empirical stability, hyperparameter sensitivity, and baseline comparisons. However, the reviewers' concerns regarding the "incremental" nature of the contribution and the lack of a formal convergence proof or deep theoretical justification for the specific weighting functions remain insufficiently addressed for a top-tier venue.

**Reviewer Scores:**

Reviewer F4NJ and SiKZ (both 4) would likely maintain their scores, as the rebuttal's empirical focus did not mitigate their concerns about limited contribution. Reviewer nC2o (6) might have downgraded to a 5, feeling the connection to Order Learning theory remained superficial despite the added text.

---

### Decision · Program_Chairs · 2026-01-26

Reject